# Kinetic Aspects of Benzene Degradation over TiO_2_-N and Composite Fe/Bi_2_WO_6_/TiO_2_-N Photocatalysts under Irradiation with Visible Light

**DOI:** 10.3390/ijms24065693

**Published:** 2023-03-16

**Authors:** Mikhail Lyulyukin, Nikita Kovalevskiy, Andrey Bukhtiyarov, Denis Kozlov, Dmitry Selishchev

**Affiliations:** 1Boreskov Institute of Catalysis, Novosibirsk 630090, Russia; lyulyukin@catalysis.ru (M.L.); nikita@catalysis.ru (N.K.); avb@catalysis.ru (A.B.); kdv@catalysis.ru (D.K.); 2Ecology and Nature Management Department, Aircraft Engineering Faculty, Novosibirsk State Technical University, Novosibirsk 630073, Russia

**Keywords:** photocatalysis, visible light, N-doped TiO_2_, Bi_2_WO_6_, composite photocatalyst, benzene degradation, thermoactivation

## Abstract

In this study, composite materials based on nanocrystalline anatase TiO_2_ doped with nitrogen and bismuth tungstate are synthesized using a hydrothermal method. All samples are tested in the oxidation of volatile organic compounds under visible light to find the correlations between their physicochemical characteristics and photocatalytic activity. The kinetic aspects are studied both in batch and continuous-flow reactors, using ethanol and benzene as test compounds. The Bi_2_WO_6_/TiO_2_-N heterostructure enhanced with Fe species efficiently utilizes visible light in the blue region and exhibits much higher activity in the degradation of ethanol vapor than pristine TiO_2_-N. However, an increased activity of Fe/Bi_2_WO_6_/TiO_2_-N can have an adverse effect in the degradation of benzene vapor. A temporary deactivation of the photocatalyst can occur at a high concentration of benzene due to the fast accumulation of non-volatile intermediates on its surface. The formed intermediates suppress the adsorption of the initial benzene and substantially increase the time required for its complete removal from the gas phase. An increase in temperature up to 140 °C makes it possible to increase the rate of the overall oxidation process, and the use of the Fe/Bi_2_WO_6_/TiO_2_-N composite improves the selectivity of oxidation compared to pristine TiO_2_-N.

## 1. Introduction

The development of advanced oxidation processes, such as photocatalytic oxidation (PCO), is permanently advanced to solve acute problems of environmental pollution and public health risk [1,2,3,4]. The PCO method has been shown to be effective in killing microorganisms and degrading hazardous chemical micropollutants in water and air environments [5,6,7]. Crystalline TiO_2_ was the primary photocatalyst in this field for a long time because it could efficiently utilize optical radiation and generate charge carriers (i.e., electrons and holes) with high enough potentials to provide degradation of contaminants in an oxygen-containing medium. A key feature of TiO_2_-mediated photocatalytic degradation is the possibility of complete oxidation of organic pollutants with the formation of harmless inorganic products such as carbon dioxide and water.

Despite the advantages, TiO_2_ has a fundamental drawback due to the large width of its band gap. For instance, the band gap of anatase TiO_2_ is 3.2 eV [8]. TiO_2_ can absorb UV radiation (<390 nm), but it cannot utilize the majority (up to 95%) of solar radiation corresponding to the visible region. This drawback promotes the development of new visible-light active photocatalysts. In addition to other narrow-band semiconductors (e.g., g-C_3_N_4_, Ag_3_PO_4_, Bi_2_MoO_6_, BiVO_4_, and ZnIn_2_S_4_ [9,10,11,12,13]), TiO_2_ having an extended action spectrum due to modifications is a promising photocatalyst for the efficient degradation of pollutants under visible light [6,14].

Surface modification of TiO_2_ with metal complexes or plasmonic metals provides a sensibilization effect and results in extending its absorption spectrum [15,16]. However, the efficiency of light absorption and generation of reactive species in this case is commonly not high enough due to the nature of electron excitation and the requirement for the transition of charge carriers to the energy bands of TiO_2_ [8]. One of the efficient methods of TiO_2_ modification is its doping with nitrogen, which results in the formation of additional energy levels in the band gap of TiO_2_ and leads to a decrease in the minimum energy required for photoexcitation of electrons. TiO_2_ doped with nitrogen (TiO_2_-N) exhibits great visible-light activity in the degradation of organic pollutants, especially, under blue light [17,18,19,20,21].

Commonly, the quantum efficiency of N-doped TiO_2_ in the visible region is not high enough due to rapid electron-hole recombination. An increase in the visible-light activity of TiO_2_-N can be achieved by the surface modification of the photocatalyst with noble or other transition metals, which play the role of containers for electrons and improve charge separation due to the creation of an energy barrier [22]. For instance, a substantial increase in the visible-light activity was observed after surface modification of TiO_2_-N with vanadium, iron, or copper species [23]. Noble metals also accelerate the transfer of electrons to adsorbed oxygen molecules, thus increasing the activity compared to that of pristine photocatalyst [24,25]. The combination of TiO_2_-N with other narrow-band semiconductors (e.g., g-C_3_N_4_, CdS, MoS_2_, Cu_2_O) can improve the separation of charge carriers due to a heterojunction, thus increasing the activity of the photocatalyst under visible light. Many composite photocatalysts based on TiO_2_-N have been described in the literature [26,27,28,29,30]. We have previously shown that Bi_2_WO_6_/TiO_2_-N heterostructure prepared by hydrothermal method exhibits a high activity in the degradation of gaseous organic pollutants under visible light and has a high stability under power radiation for a long time [31]. Similar to TiO_2_-N, the visible-light activity of the Bi_2_WO_6_/TiO_2_-N composite can be substantially increased by depositing metal species (e.g., Fe) on its surface [32]. All three types of photocatalysts, namely TiO_2_-N, Bi_2_WO_6_/TiO_2_-N, and Fe/Bi_2_WO_6_/TiO_2_-N, were selected as the main objects of this study to illustrate the effect of the photocatalyst’s ability on the kinetic aspects of degradation of volatile organic compounds. The study is focused on benzene as a representative of aromatic compounds because this class of pollutants has a major harmful impact on human health. Analysis of pollutant adsorption and formation of intermediates and final degradation products is important for identification of the reaction pathways because they can change for heterostructures due to a change in the potentials of charge carriers photogenerated under visible light. Furthermore, the degradation of different classes of organic pollutants can occur through different pathways that would affect the overall reaction rate. To illustrate this statement in the paper, we discuss the kinetic aspects of the degradation of ethanol and benzene as nonaromatic and aromatic compounds, respectively. The kinetics of adsorption and degradation of both pollutants is studied in detail in a batch reactor using in situ IR spectroscopy. We show that under certain operating conditions the high photocatalytic ability of a material can have an adverse effect on the removal of aromatic pollutants from the air.

## 2. Results and Discussion

### 2.1. Characteristics of Synthesized Photocatalysts

Calcination of the precipitate formed after mixing aqueous solutions of titanium oxysulfate and ammonia results in the crystallization of anatase nanoparticles (Figure 1a). The main peaks at 2θ of 25.3°, 37.8°, 48.1°, 53.9°, 55.1°, and 62.8° correspond to the (101), (004), (200), (105), (211), and (204) planes of anatase TiO_2_ (JCPDS card No.21-1272). The average size of anatase crystallites was estimated from the XRD data to be 20 nm. The formation of the rutile phase is not observed even under high-temperature treatment.

Due to using ammonia as a precipitation agent, some nitrogen species are incorporated into the crystal structure of the formed TiO_2_ [32,33,34]. According to the results of CHNS analysis, the total content of nitrogen in the solid titania sample after calcination in air at 450 °C is 0.36 ± 0.03 wt.%, which corresponds to a high value. The peak with a BE of 399.9 eV was observed in the photoelectron N1s spectral region of as-prepared TiO_2_-N (Figure 2a). This peak can be attributed to a form of partially oxidized nitrogen located in the interstitial positions of the TiO_2_ lattice [35]. The presence of nitrogen species was also confirmed by the EPR technique, which showed a signal of paramagnetic N centers (Appendix A). According to the results of XPS analysis, titanium in the prepared TiO_2_-N is present in the charge state of Ti^4+^ (Figure 2b).

The nitrogen species create additional energy levels in the band gap of TiO_2_, thus decreasing the minimum energy required for photoexcitation of electrons to the conduction band [36]. Figure 3a illustrates this statement and shows that TiO_2_-N has an additional absorption shoulder in the range of 400–540 nm with a maximum at ca. 450 nm. According to the Tauc plot, shown in the inset in Figure 3b, the minimum excitation energy for TiO_2_-N can be estimated to be 2.3 eV, which is substantially lower than the band gap of anatase (3.2 eV, Figure 3b). Therefore, TiO_2_-N absorbs visible light with wavelengths up to 540 nm (mainly in the blue region). We have previously shown that the as-prepared TiO_2_-N photocatalyst exhibits a high photocatalytic ability in the degradation of volatile organic compounds under both UV and visible light [32].

Hydrothermal synthesis using aqueous solutions of Bi(NO_3_)_3_ and Na_2_WO_4_ results in the formation of orthorhombic Bi_2_WO_6_ with a lamellar structure presented in the form of nanoplates (Figure 1c). The main reflection peaks in the XRD pattern at 2Θ of 28.3°, 32.9°, 47.2°, 55.9°, and 58.6° correspond to the (131), (200), (202), (133), and (262) planes of orthorhombic Bi_2_WO_6_ (PDF 39-0256). XPS analysis confirms the chemical composition of the prepared Bi_2_WO_6_ sample because the corresponding peaks in the spectral regions of Bi4f and W4f (Figure 2c,d) are unambiguously attributed to the charge states of +3 for Bi and +6 for W [37]. The absence of other states for these elements confirms the formation of single-phase material without non-stochiometric species.

As a typical Aurivillius oxide, Bi_2_WO_6_ is a semiconducting material with a band gap narrower than that of anatase TiO_2_. Figure 3a shows that Bi_2_WO_6_ can absorb a small portion of photons in the visible region (up to 440 nm) because its optical band gap corresponds to a value of 2.8 eV (Figure 3b). On the other hand, the absorption of visible light by Bi_2_WO_6_ is much weaker if compared with TiO_2_-N material. Due to this reason, the optical properties of the Bi_2_WO_6_/TiO_2_-N composite based on both materials (Figure 3a) are similar to the properties of single TiO_2_-N. Figure 1b shows that titania nanoparticles in the Bi_2_WO_6_/TiO_2_-N composite cover the external surface of agglomerated Bi_2_WO_6_ nanoplates and provide strong absorption of visible light. More illustrations of local composite structure can be found in Appendix A, which shows TEM micrographs of Bi_2_WO_6_/TiO_2_-N obtained using HAADF imaging and EDX mapping techniques.

An approximation of the Tauc plot attributed to Bi_2_WO_6_/TiO_2_-N sample (Figure 3b) gives the values of 3.17 eV as the energy of band-to-band excitation of electrons in anatase and 2.33 eV as the minimum energy required for excitation of electrons in this system due to the presence of nitrogen energy levels in the band gap of anatase.

Similarly to a single Bi_2_WO_6_, XPS analysis of Bi_2_WO_6_/TiO_2_-N shows only the charge states of +3 and +6 for Bi and W elements, respectively (Figure 2e,f). No other states of these elements, as well as no change in the charge states of Ti and N (see Appendix A), confirm no doping the TiO_2_ lattice with Bi or W species under hydrothermal treatment.

The realization of heterojunction in Bi_2_WO_6_/TiO_2_-N leads to a substantial increase in the visible-light activity of this composite compared to initial TiO_2_-N [31]. For instance, the steady-state rate of CO_2_ formation during the oxidation of acetone vapor under blue light (450 nm, 160 mW cm^−2^) was 0.7 and 1.0 µmol min^−1^ for TiO_2_-N and Bi_2_WO_6_/TiO_2_-N, respectively. According to the literature data [38,39], the energy bands of Bi_2_WO_6_ are located lower than the bands of TiO_2_ that makes possible a type II heterojunction between the semiconductors or a Z-scheme heterojunction. Both types of heterojunctions can improve the separation of photogenerated charge carriers and suppress their recombination, thus leading to an increase in the visible-light activity.

As mentioned in the Introduction, the photocatalytic ability of Bi_2_WO_6_/TiO_2_-N can be improved by deposing a small amount of Fe species on its surface. In the Fe-modified photocatalysts, localization of photogenerated electrons occurs on the surface of the photocatalyst in Fe species, and the separation of charge carriers is enhanced due to an energy barrier. Bi_2_WO_6_/TiO_2_-N was impregnated with an aqueous solution of Fe(NO_3_)_3_ to achieve a nominal Fe loading of 0.1 wt.%. This modification had no effect on the XRD pattern of Bi_2_WO_6_/TiO_2_-N due to extremely low Fe content. The presence of Fe species on the surface of the photocatalyst was confirmed by XPS analysis when the Fe content was increased up to 0.3 wt.% (see Appendix A).

The photocatalytic ability of all synthesized catalysts was preliminary checked in a continuous-flow setup as the steady-state rate of CO_2_ formation during the oxidation of acetone vapor under blue light (450 nm, 160 mW cm^−2^). The ability is increased as follows: TiO_2_-N:Bi_2_WO_6_/TiO_2_-N:Fe/Bi_2_WO_6_/TiO_2_-N = 1:1.4:1.7. The next section describes the effect of photocatalyst ability on the kinetic aspects of the degradation of different classes of organic compounds.

### 2.2. Degradation of Volatile Organic Compounds

In this study, we investigate the photocatalytic degradation of ethanol and benzene vapor on the surface of TiO_2_-N, Bi_2_WO_6_/TiO_2_-N, and Fe/Bi_2_WO_6_/TiO_2_-N photocatalysts, which substantially differ in their activities. These test organic compounds were selected due to the different types of intermediates formed during the degradation because they can affect the deactivation of the photocatalyst and, consequently, the kinetics of overall oxidation process [40,41]:The PCO of ethanol results in the formation of products that are easily desorbed from the surface of photocatalyst (especially acetaldehyde, which is the first in the sequence of oxidation products) [42,43];The PCO of benzene occurs without the formation of gas-phase intermediates but leads to strongly adsorbed species formed on the surface of the photocatalyst due to the polymerization of intermediate radicals [44,45,46]. Furthermore, oxidation pathways with the opening of the benzene ring lead to an increase in the total number of species, thus resulting in the gradual deactivation of the photocatalyst due to competition for adsorption sites with the initial compound [47,48].

A common pattern for the photocatalytic experiments in a batch reactor is that the injected liquid pollutant evaporates and partially adsorbs on the surface of the photocatalyst. Turning on the light leads to a decrease in the concentration of pollutant in the gas phase and a simultaneous increase in the amount of oxidation products. The intermediate oxidation products (if any) are completely removed from the gas phase during irradiation, while carbon oxides start to intensively release. The total amount of carbon oxides is commonly mass-balanced to the initial amount of compound injected into the reactor. It confirms the complete oxidation of the injected pollutant.

#### 2.2.1. Degradation of Ethanol in Batch Reactor

Figure 4 shows the kinetic plots of the ethanol PCO over all samples: TiO_2_-N, Bi_2_WO_6_/TiO_2_-N, and Fe/Bi_2_WO_6_/TiO_2_-N. The initial amount of ethanol injected into the reactor (1 µL) corresponds to its concentration of 57 µmol L^−1^, but when the adsorption–desorption equilibrium is reached, the detected concentration of ethanol in the gas phase is 8–10 µmol L^−1^. This fact is due to the high adsorption capacity of synthesized TiO_2_-based photocatalyst toward ethanol. Complete removal of ethanol vapor from the gas phase during the irradiation of the photocatalyst occurs for 45–55 min both for TiO_2_-N and Bi_2_WO_6_/TiO_2_-N (Figure 4). In the case of Fe/Bi_2_WO_6_/TiO_2_-N, the complete removal of ethanol occurs for 20 min.

At the initial steps of ethanol oxidation, acetaldehyde is formed as a gas-phase intermediate. The rates of acetaldehyde formation are 0.95, 1.5, and 2.05 µmol L^−1^ min^−1^ for TiO_2_-N, Bi_2_WO_6_/TiO_2_-N, and Fe/Bi_2_WO_6_/TiO_2_-N, respectively. Acetaldehyde is adsorbed less strongly than ethanol [42], thus it does not participate in the redox transformations on the surface of the photocatalyst while the adsorption sites are occupied by ethanol molecules. Oxidation of acetaldehyde boosts after the complete consumption of ethanol and an increase in vacant adsorption sites on the surface of the photocatalyst. Therefore, the time of maximum acetaldehyde concentration coincides with the time of complete removal of ethanol from the gas phase. For all studied photocatalysts, the maximum concentration of acetaldehyde detected in the gas phase is ca. 18 µmol L^−1^. Figure 4 shows that the complete removal of acetaldehyde from the gas phase occurs for 100, 77, and 50 min for TiO_2_-N, Bi_2_WO_6_/TiO_2_-N, and Fe/Bi_2_WO_6_/TiO_2_-N, respectively. In the case of TiO_2_-N, formic acid (<0.45 µmol L^−1^) is also detected in the gas phase. Furthermore, 1.6 µmol L^−1^ of formaldehyde and 0.5 µmol L^−1^ of formic acid are detected during the oxidation of ethanol over Bi_2_WO_6_/TiO_2_-N, whereas Fe/Bi_2_WO_6_/TiO_2_-N leads to the formation of formaldehyde in the concentration of 2 µmol L^−1^.

CO_2_ plots in Figure 4 show that the times required for complete conversion of ethanol to carbon oxides are 320, 180, and 120 min for TiO_2_-N, Bi_2_WO_6_/TiO_2_-N, and Fe/Bi_2_WO_6_/TiO_2_-N, respectively, that corresponds to maximum CO_2_ formation rate of 0.5, 0.68, and 1.4 µmol L^−1^ min^−1^. This trend agrees with the results of acetone degradation under blue light, which are discussed in Section 2.1.

Additionally, the effect of the basic wavelength in the emission spectrum of LED on the kinetics of ethanol oxidation over TiO_2_-N was studied to understand the reason for the formation of gaseous formaldehyde in the case of composite photocatalysts. Appendix A shows that the rate of ethanol oxidation increases as the wavelength decreases, but the release of formaldehyde in the gas phase is not observed for all the experiments. This means that an increase in the activity of the photocatalyst increases the rates of all steps. No additional pathways with an intense formation of formaldehyde appear for the TiO_2_-N photocatalyst. Therefore, the release of formaldehyde in the case of composite photocatalysts occurs due to the presence of the Bi_2_WO_6_ co-catalyst. In fact, gaseous formaldehyde is detected during the oxidation of ethanol vapor over Bi_2_WO_6_ (see Appendix A). The low concentration of formaldehyde in this case is due to the very low activity of the Bi_2_WO_6_ photocatalyst. A high amount of formaldehyde in the case of and Bi_2_WO_6_/TiO_2_-N composites may be due to a change in the potentials of photogenerated charge carriers and a difference in the oxidation rates over TiO_2_-N and Bi_2_WO_6_ components.

#### 2.2.2. Degradation of Benzene in Batch Reactor

The kinetic plots of the reaction components during the oxidation of benzene vapor over TiO_2_-N, Bi_2_WO_6_/TiO_2_-N, and Fe/Bi_2_WO_6_/TiO_2_-N under irradiation with blue light are shown in Figure 5. As TiO_2_-N poorly adsorbs benzene molecules on its surface, the concentration of benzene detected in the gas phase after evaporation is 37 µmol L^−1^ that corresponds to the theoretical value calculated from the amount of injected benzene. For the composite Bi_2_WO_6_/TiO_2_-N and Fe/Bi_2_WO_6_/TiO_2_-N photocatalysts, the starting concentration of benzene in the gas phase is 33–35 µmol L^−1^ due to the adsorption of benzene molecules on the surface of Bi_2_WO_6_. After turning the light source on, complete removal of benzene from the gas phase occurs after 380, 610, and 700 min for TiO_2_-N, Bi_2_WO_6_/TiO_2_-N, and Fe/Bi_2_WO_6_/TiO_2_-N, respectively. We can see the opposite trend compared to the previous cases of acetone and ethanol degradation. The reasons for this trend will be discussed later.

The accumulation of CO and CO_2_ occurs simultaneously with the degradation of benzene (Figure 5). After the complete removal of benzene from the gas phase, the formation of final products is substantially enhanced. The rates of CO_2_ formation for TiO_2_-N are 0.35 µmol L^−1^ min^−1^ before the complete removal of benzene and 0.54 µmol L^−1^ min^−1^ after that. The corresponding values for Bi_2_WO_6_/TiO_2_-N and Fe/Bi_2_WO_6_/TiO_2_-N are 0.19 (and 0.55) µmol L^−1^ min^−1^ and 0.17 (and 0.82) µmol L^−1^ min^−1^, respectively. These data show that the higher ability of photocatalyst leads to the lower rate of CO_2_ formation at initial steps of reaction. After complete removal of benzene from the gas phase, the higher photocatalytic ability provides the higher rate of CO_2_ accumulation. The reasons for the observed phenomenon may be related to the mechanism of benzene oxidation.

According to the literature data [44,45,46], the holes (h^+^) photogenerated in the conduction band of the photocatalyst can directly react with adsorbed benzene molecules to form phenyl radical cations (C_6_H_5_^•+^), as well as oxidize adsorbed H_2_O molecules to form OH^•^ radicals. C_6_H_5_^•+^ can further react with the adsorbed O_2_ or O_2_^•−^ to form a peroxide radical and, sequentially, phenol [49]. An alternative pathway involves the direct interaction of C_6_H_5_^•+^ with OH^•^ to form phenol and other hydroxylated intermediates (e.g., hydroquinone and benzoquinone) [50]. These compounds can be completely oxidized to CO_2_ and H_2_O through a series of steps that include opening the aromatic ring and oxidation of non-cyclic hydrocarbons. In parallel pathways, C_6_H_5_^•+^ can interact with other adsorbed benzene molecules to form carbon deposits due to polymerization. Accumulation of these deposits can substantially reduce the ability of photocatalyst to oxidize benzene molecules due to blocking its adsorption sites. Other transformation routes [51] include the reactions of photoinduced OH^•^ with adsorbed benzene molecules to form various types of alkyl radicals (e.g., CH_3_^•^, C_2_H_5_^•^ and C_6_H_5_^•^), which also contribute to the overall decomposition of benzene [52].

Einaga et al. [45,46] previously showed using diffuse reflectance spectroscopy and other experimental techniques that a radical polymerization of aromatic compounds on the surface of TiO_2_ during the degradation of benzene vapor leads to the formation of carbon deposits on the surface of photocatalyst and its strong deactivation. Indirect evidence for the formation of non-volatile intermediates in this study is that benzene is poorly adsorbed on the surface of the studied photocatalysts but ~30% of the formed CO_2_ appears in the gas phase only after complete removal of benzene vapor. Therefore, it can be concluded that the non-linear form of the CO_2_ kinetic plots in Figure 5 is due to the accumulation of non-volatile intermediates, which occupy available sites on the surface of photocatalyst and suppress further redox transformations.

To confirm the adverse effect of an excessive amount of benzene on the kinetics of its degradation, an additional experiment with 0.15 µL of injected benzene, instead of 1.0 µL, was carried out over TiO_2_-N and Fe/Bi_2_WO_6_/TiO_2_-N under the same conditions (Figure 6). When oxidizing 0.15 µL of benzene over TiO_2_-N, the previously observed stage of CO_2_ formation at a lower rate is absent, and the rate of the second (“fast”) stage is the same as in the case of 1.0 µL (i.e., 0.55 µmol L^−1^ min^−1^). When the same amount of benzene is oxidized over the Fe/Bi_2_WO_6_/TiO_2_-N composite, the slope at the initial stage is 0.34 µmol L^−1^ min^−1^, which is twice as high as the rate observed during the oxidation of 1 µL (Figure 5). This means that the coverage of photocatalyst surface has a drastic effect on the rate of benzene oxidation because non-volatile intermediates remain on the surface and can lead to a fast transfer and recombination of the photogenerated charge carriers [44]. As the adsorption of benzene on the composite Fe/Bi_2_WO_6_/TiO_2_-N photocatalyst is slightly better and its photocatalytic ability is higher, a higher number of non-volatile species is formed on its surface at the initial steps, thus suppressing the degradation of benzene. After complete removal of benzene from the gas phase, the number of intermediates on the photocatalyst surface decreases, and the accumulation of final products increases. The rate of CO_2_ accumulation during the “fast” oxidation stage for Fe/Bi_2_WO_6_/TiO_2_-N (Figure 6) is ca. 1 µmol L^−1^ min^−1^, which is substantially higher than the value observed during the oxidation of 1 µL of benzene.

Based on the results of all experiments performed in the batch reactor, it can be concluded that a multicomponent composite Fe/Bi_2_WO_6_/TiO_2_-N system leads to an increase in the photocatalytic ability of the material compared to pristine TiO_2_-N (Figure 6). However, the oxidation of aromatic compounds using this system may be associated with limitations because the high ability of photocatalyst promotes the occupation of available adsorption sites by non-volatile intermediates and its temporary deactivation. An illustration of the proposed deactivation mechanism by non-volatile intermediates for two types of photocatalysts (i.e., low-active and high-active photocatalysts) is shown in Figure 7.

In contrast to the oxidation of ethanol vapor, in which the intermediates are desorbed from the surface of photocatalyst to the gas phase, the intermediates formed during the benzene degradation strongly suppress the adsorption of benzene molecules and, consequently, the rate of the overall oxidation process, estimated as the accumulation of final products. In real operational conditions, the concentration of benzene is rarely as high as in the experiments described above. The concentration would be at the level of its threshold limit value (TLV = 0.1 µmol L^−1^), which is 400 times less than the amount used in the experiments. The results show that the visible-light oxidation of even 5.5 µmol L^−1^ of benzene, which exceeds the TLV by 55 times, occurs faster over Fe/Bi_2_WO_6_/TiO_2_-N compared to pristine TiO_2_-N. We believe that the problem of temporary photocatalyst deactivation can be solved by the design of a composite system with highly porous support (for example, activated carbon or zeolite), which is able to absorb a large amount of aromatic compounds and reduce their maximum concentration in the reaction mixture [53].

#### 2.2.3. Degradation of Benzene in Continuous-Flow Reactor

The experiments in a batch reactor thoroughly simulate a situation of fast emission of a high amount of pollutant to the gas phase. However, a common case is the presence of an emission source that emits a small amount of pollutant at a constant rate (e.g., pieces of furniture can slowly emit organic compounds). The experiments under flow conditions are important to check the photocatalytic performance of materials for this case too. In contrast to a batch reactor, where concentrations of reaction components are changed for irradiation time, in a continuous-flow set-up, the concentrations of initial pollutant and formed products commonly reach steady-state values, which can be used for evaluation of the steady-state activity of photocatalyst. Additionally, the experiments under flow conditions allow us to check the long-term stability of photocatalysts under irradiation.

Figure 8 shows a change in the rate of CO_2_ formation (µmol min^−1^) during long-term benzene degradation in a continuous-flow setup over TiO_2_-N and Fe/Bi_2_WO_6_/TiO_2_-N irradiated with blue light. The results of these experiments confirm a high stability of both synthesized photocatalysts because their activities reach a steady-state level after irradiation for 10–12 h and further do not change substantially up to 26 h. It is important to note that both synthesized photocatalysts exhibit a high level of activity because the activity of commercially available TiO_2_ P25 photocatalyst under the same conditions is 0.033 µmol min^−1^, that is 10 times lower compared to the values for TiO_2_-N and Fe/Bi_2_WO_6_/TiO_2_-N (see Appendix A).

The surface composition of the photocatalysts after long-term benzene degradation was checked using XPS technique. No change in the positions of peaks attributed to the main elements is detected in the XPS spectra of Fe/Bi_2_WO_6_/TiO_2_-N before and after the long-term stability test (Figure 9). The same situation is observed for the single TiO_2_-N sample (see Appendix A). Additionally, Table 1 shows that the atomic ratios of the main elements on the surface of Fe/Bi_2_WO_6_/TiO_2_-N are similar before and after the stability test that confirms a stable chemical composition of this photocatalyst.

Therefore, the kinetic data and physicochemical characteristics confirm that the synthesized photocatalysts are stable and can be successfully employed in the photocatalytic air purifiers for permanent removal and degradation of benzene micropollutants.

Another important parameter that affects the reaction rate is the temperature of the photocatalyst. The published information on the thermal activation of photocatalytic oxidation suggests some potential advantages for accelerating the process at high temperatures [54]. It was a reason to evaluate the steady-state activity of the synthesized photocatalysts at different temperatures from 40 ℃ to 140 ℃. The results of these experiments for all studied photocatalysts are shown in Figure 10.

Several statements can be made considering the observed temperature dependencies:The activity of all samples in benzene degradation increases as the reaction temperature increases, even after overcoming a value of 80 °C, which is commonly regarded as the inflection point in the oxidation processes [55], because “the exothermic adsorption of reactant becomes disfavored and tends to become the rate limiting-step”. This means that there is no actual limitation in the thermoactivation of benzene oxidation because benzene is poorly adsorbed on the photocatalyst surface.Under the conditions of this experiment, TiO_2_-N is more active than the composite Bi_2_WO_6_/TiO_2_-N and Fe/Bi_2_WO_6_/TiO_2_-N photocatalysts over the whole temperature range (the reasons for that are discussed in the previous section). However, if we look at the formation rates of both final products: CO_2_ as the product of complete oxidation (Figure 10a) and CO as the product of incomplete oxidation (Figure 10b), the Fe/Bi_2_WO_6_/TiO_2_–N photocatalyst gives a lower amount of CO and provides more selective oxidation of benzene compared to other samples. This means that this photocatalyst would more effectively reduce the total hazard of air polluted with benzene vapor.


## 3. Materials and Methods

High purity grade titanium(IV) oxysulfate (TiOSO_4_), bismuth(III) nitrate pentahydrate (Bi(NO_3_)_3_·5H_2_O), and sodium tungstate dihydrate (Na_2_WO_4_·2H_2_O) were purchased from Sigma-Aldrich (St. Louis, MO, USA). Reagent grade ammonium hydroxide solution (NH_4_OH, 25%) and nitric acid (HNO_3_, 65%) were purchased from AO Reachem Inc. (Moscow, Russia). These reagents were used for the synthesis of photocatalysts as received without further purification. N-doped TiO_2_ (TiO_2_-N) was synthesized under neutral conditions by precipitation from TiOSO_4_ aqueous solution using ammonia followed by washing of the precipitate with deionized water and its calcination in air at 450 ℃. Parameters of synthesis were selected based on the results of our previous study [32] to prepare TiO_2_-N with a high activity under visible light. Bi_2_WO_6_/TiO_2_-N composite was synthesized by hydrothermal method using Bi(NO_3_)_3_ and Na_2_WO_4_ precursors according to our previously published technique [31]. A molar ratio between Bi_2_WO_6_ and TiO_2_-N components (5:100) was adjusted by the initial amounts of precursors and TiO_2_-N in suspension. The Bi_2_WO_6_/TiO_2_-N composite was modified by depositing Fe species on its surface using an aqueous solution of iron nitrate. The estimated content of Fe was 0.1 wt.%. This sample is denoted in the paper as Fe/Bi_2_WO_6_/TiO_2_-N.

The total content of nitrogen was measured by CHNS analysis using a Vario EL Cube elemental analyzer from Elementar Analysensysteme GmbH (Langenselbold, Germany). The crystal phases in the prepared photocatalysts were analyzed using powder X-ray diffraction (XRD). The data were collected in the 2θ range of 10–75° with a step of 0.05° and a collection time of 3 s using a D8 Advance diffractometer (Bruker, Billerica, MA, USA) equipped with a CuKα radiation source and a LynxEye position sensitive detector. SEM micrographs were received using an ultra-high-resolution Field-Emission SEM (FE-SEM) Regulus 8230 (Hitachi, Tokyo, Japan) at an accelerating voltage of 5 kV. X-ray photoelectron spectroscopy (XPS) analysis was performed using a SPECS photoelectron spectrometer (SPECS Surface Nano Analysis GmbH, Berlin, Germany) equipped with a PHOIBOS-150 hemi-spherical energy analyzer and an AlKα radiation source (hν = 1486.6 eV, 150 W). The binding energies (BE) were pre-calibrated using the lines of Au4f7/2 (84.0 eV) and Cu2p3/2 (932.67 eV) from metallic gold and copper foils. Peak fitting in the collected spectra was performed using XPSPeak 4.1 software (Informer Technologies Inc., Los Angeles, CA, USA). UV-Vis diffuse reflectance spectroscopy was used to examine the optical properties of the samples at room temperature. The spectra were taken on a Cary 300 UV-Vis spectrophotometer from Agilent Technologies Inc. (Santa Clara, CA, USA) equipped with a DRA-30I diffuse reflectance accessory and special pre-packed polytetrafluoroethylene as a reflectance standard in the range of 250–850 nm. The optical band gap was calculated using the Tauc method, assuming indirect allowed excitations.

The prepared photocatalysts were tested in the degradation of ethanol and benzene vapors under blue light. The kinetic aspects were studied at 25 °C in a 0.3 L batch reactor installed in the cell compartment of a Vector 22 FTIR spectrometer (Bruker, Billerica, MA, USA). The details of the experimental setup can be found elsewhere [56]. A sample of photocatalyst was uniformly deposited onto a 9 cm^2^ glass support to obtain an even layer with an area density of 5 mg cm^−2^. Then, this support was placed into the reactor with a quartz window. The reactor was purged for 45 min with purified air with a relative humidity of 20% and tightly closed. After that, the photocatalyst was irradiated for 30 min using a light-emitting diode (blue LED), which had a maximum in its emission spectrum at 445 nm (Figure 11a). The photon flux to the surface of photocatalyst provided by the blue LED was 120 µE min^−1^. LED was turned off after this pretreatment required for the oxidation of all the organic species previously adsorbed on the surface of photocatalyst. Then, 1 µL of liquid ethanol or benzene was injected into the reactor, and IR spectra of the gas phase were collected periodically to analyze volatile compounds in the reactor. The blue LED was turned on again after 20–30 min when the adsorption–desorption equilibrium was reached. The irradiation continued until the total concentration of the C-containing products accumulated in the gas phase reached the theoretically estimated level. The starting point (i.e., 0 min) in all kinetic plots was placed to the moment of light on.

In addition to the experiments in the batch reactor, stability of photocatalysts and their steady-state activity were studied in a continuous-flow set-up operated under conditions as follows: the volume flow rate was 0.10 ± 0.02 L min^−1^; the relative humidity of inlet air was 20%, the inlet concentration of benzene was 1–12 μmol L^−1^, the temperature of photoreactor was 40–140 °C. The details of this continuous-flow set-up can be found elsewhere [54]. The photocatalyst was irradiated with a blue LED, which had a maximum in its emission spectrum at 441 nm (Figure 11b). The photon flux to the surface of the photocatalyst in this case was 270 µE min^−1^. The steady-state rate of CO_2_ formation (µmol min^−1^) during benzene degradation was evaluated to compare the activity of photocatalysts. The results of preliminary experiments (see Appendix A) showed that the rate of CO_2_ formation for all studied photocatalysts slightly depends on the concentration of benzene in the region higher than 2 µmol L^−1^. Therefore, the initial concentration of benzene was adjusted to 10 ± 0.5 µmol L^−1^ for the correct comparison of different photocatalysts. The temperature of photoreactor was varied from 40 °C to 140 °C to study its effect on the activity.

## 4. Conclusions

Photocatalysts based on N-doped TiO_2_ (TiO_2_-N) exhibit high activity in the degradation of volatile organic compounds under visible light. Ethanol and benzene are used as examples of non-aromatic and aromatic compounds, respectively, to study the kinetic aspects of the photocatalytic degradation. A combination of TiO_2_-N with bismuth tungstate in a composite system (Bi_2_WO_6_/TiO_2_-N) and subsequent modification of its surface with iron species (Fe/Bi_2_WO_6_/TiO_2_-N) substantially increase the removal rate of the initial pollutant and formed intermediates from the gas phase compared to pristine TiO_2_-N. However, the high photocatalytic ability of the composite system can have an adverse effect in the case of benzene because the rate of pollutant degradation is substantially decreased compared to a pristine TiO_2_-N photocatalyst, especially at high concentrations. This effect occurs due to a fast accumulation of non-volatile intermediates, which occupy available adsorption sites on the surface of the photocatalyst, thus suppressing the adsorption of initial benzene and, consequently, reducing the rate of the overall oxidation process. The observed deactivation is reversible. When all benzene is removed from the gas phase, the surface of the photocatalyst is rapidly cleaned with the formation of a high amount of CO_2_ in the gas phase. The rate of benzene degradation is monotonically increased as the reaction temperature is increased up to 140 °C. At the same time, this thermal activation does not increase the relative yield of CO as a by-product that allows a fast and efficient reduction in the total hazard of air polluted with benzene vapor.

## Figures and Tables

**Figure 1 ijms-24-05693-f001:**
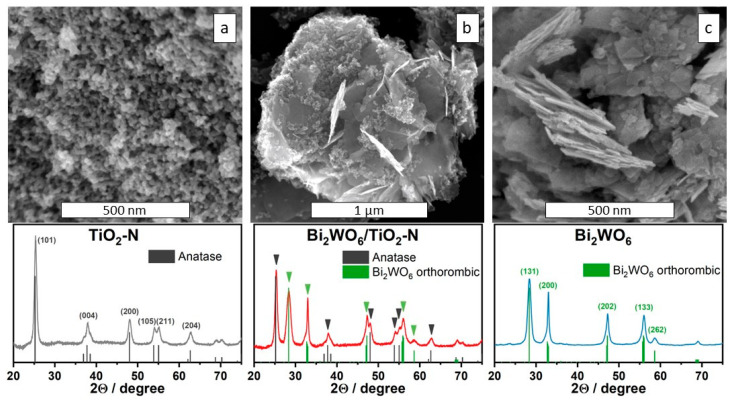
SEM micrographs and XRD patterns of synthesized TiO_2_-N (**a**), Bi_2_WO_6_/TiO_2_-N (**b**), and Bi_2_WO_6_ (**c**) photocatalysts.

**Figure 2 ijms-24-05693-f002:**
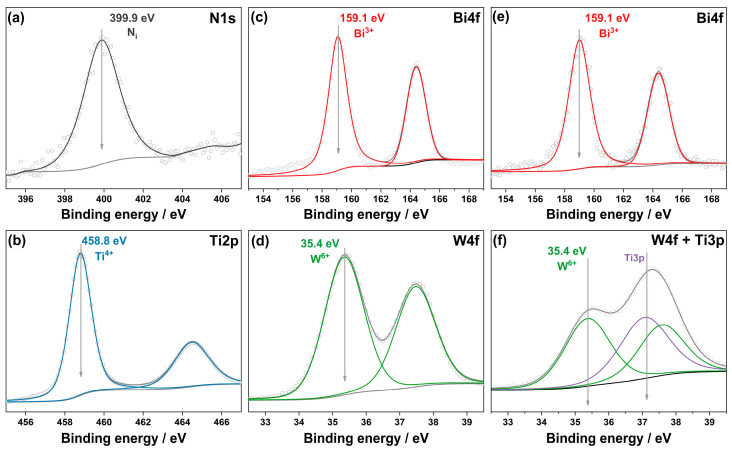
Photoelectron N1s (**a**) and Ti2p (**b**), Bi4f (**c**) and W4f (**d**), Bi4f (**e**) and W4f + Ti3p (**f**) spectral regions for TiO_2_-N, Bi_2_WO_6_, and Bi_2_WO_6_/TiO_2_-N samples, respectively. The experimental data points are marked with gray circles, the fitted data and Shirley-type backgrounds are shown with lines.

**Figure 3 ijms-24-05693-f003:**
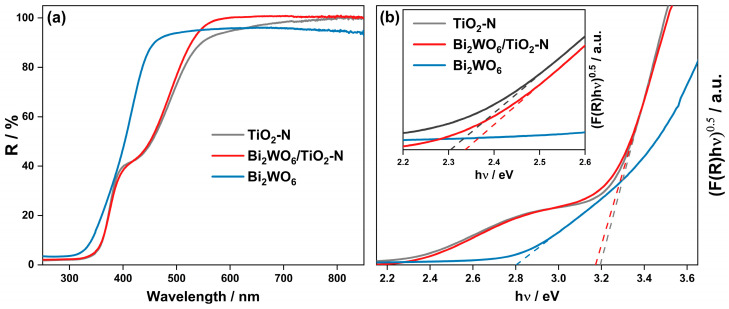
UV–vis diffuse reflectance spectra of TiO_2_-N, Bi_2_WO_6_/TiO_2_-N, and Bi_2_WO_6_ samples (**a**) and the corresponding Tauc plots (**b**). The inset in figure (**b**) shows an enlarged area of Tauc plots in the range of 2.2–2.6 eV.

**Figure 4 ijms-24-05693-f004:**
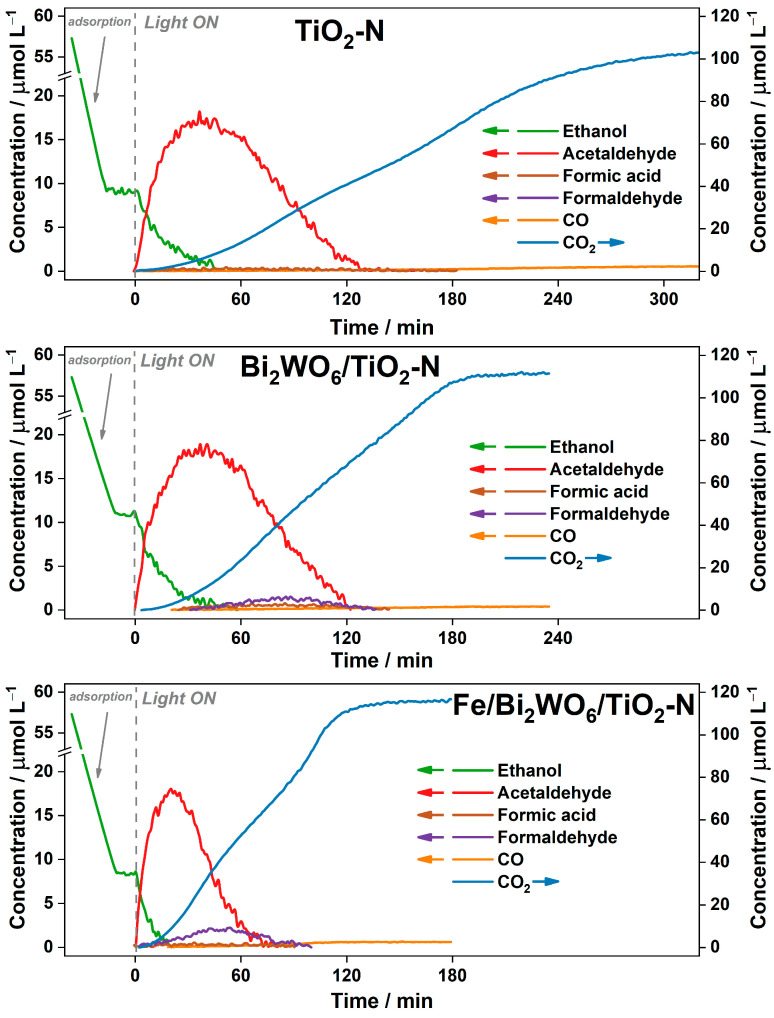
Kinetic plots of the reaction components during the PCO of ethanol over TiO_2_-N, Bi_2_WO_6_/TiO_2_-N, and Fe/Bi_2_WO_6_/TiO_2_-N under blue light.

**Figure 5 ijms-24-05693-f005:**
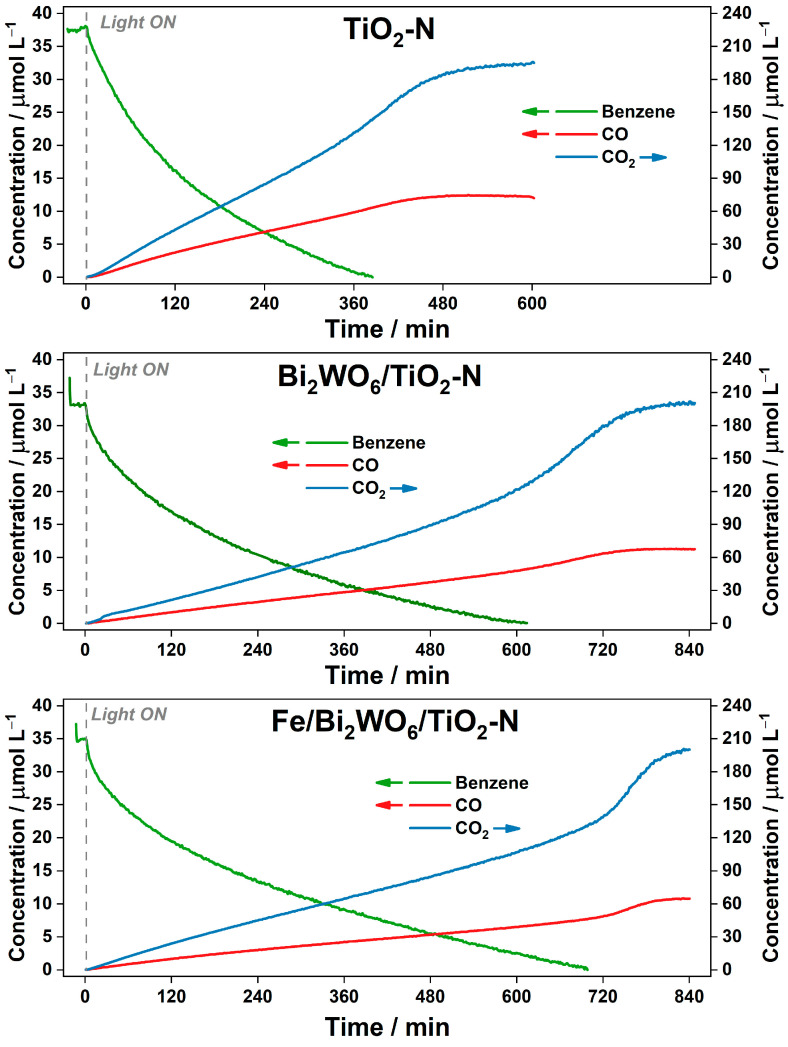
Kinetic plots of the reaction components during PCO of benzene (1.0 µL) over TiO_2_-N, Bi_2_WO_6_/TiO_2_-N, and Fe/Bi_2_WO_6_/TiO_2_-N under blue light.

**Figure 6 ijms-24-05693-f006:**
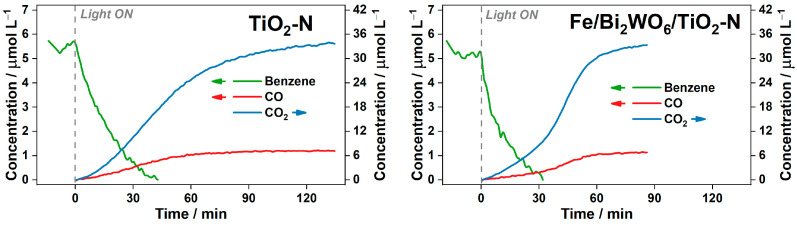
Kinetic plot of the reaction components during PCO of benzene (0.15 µL) over TiO_2_-N and Fe/Bi_2_WO_6_/TiO_2_-N under blue light.

**Figure 7 ijms-24-05693-f007:**
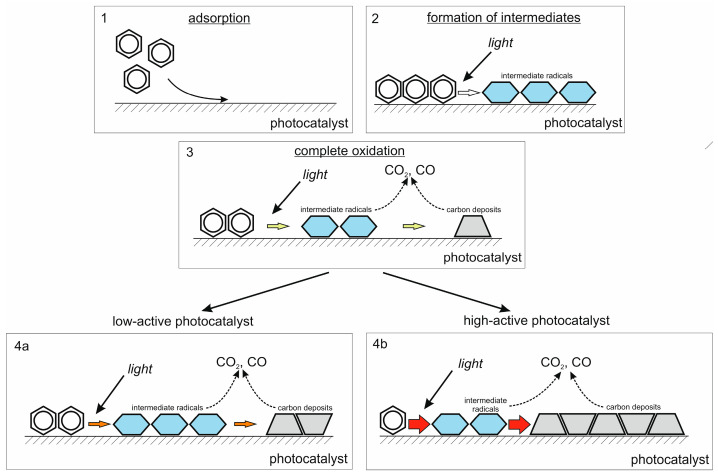
Proposed scheme of photocatalyst deactivation during the benzene degradation as a series of steps: 1—adsorption of benzene; 2—formation of intermediate oxidation products; 3—overall oxidation process, including the formation of final products; 4a and 4b—complete oxidation over low-active and high-active photocatalysts, respectively.

**Figure 8 ijms-24-05693-f008:**
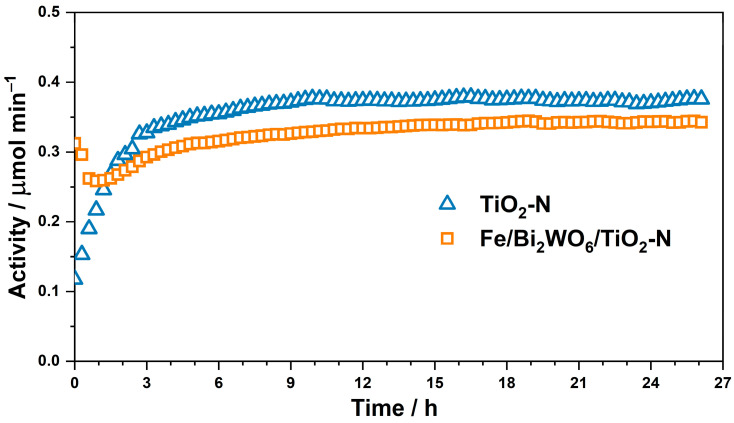
Stability of TiO_2_-N and Fe/Bi_2_WO_6_/TiO_2_-N during long-term benzene degradation under blue light.

**Figure 9 ijms-24-05693-f009:**
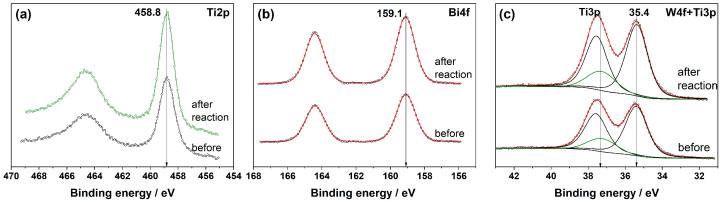
Photoelectron Ti2p (**a**), Bi4f (**b**), and W4f + Ti3p (**c**) spectral regions for the Fe/Bi_2_WO_6_/TiO_2_-N photocatalyst before and after the stability test.

**Figure 10 ijms-24-05693-f010:**
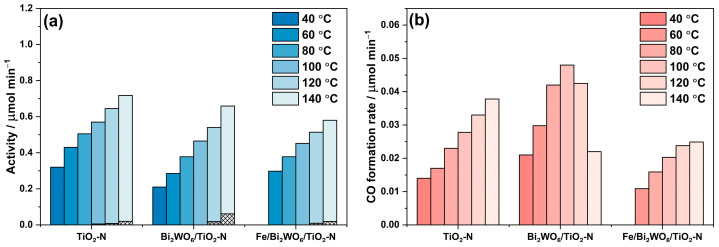
Effect of temperature on the steady-state photocatalytic activities of TiO_2_-N, Bi_2_WO_6_/TiO_2_-N, and Fe/Bi_2_WO_6_/TiO_2_-N (**a**) and CO formation rates (**b**) during the benzene degradation under blue light. Cross-hatched area corresponds to the activity without irradiation of photocatalyst.

**Figure 11 ijms-24-05693-f011:**
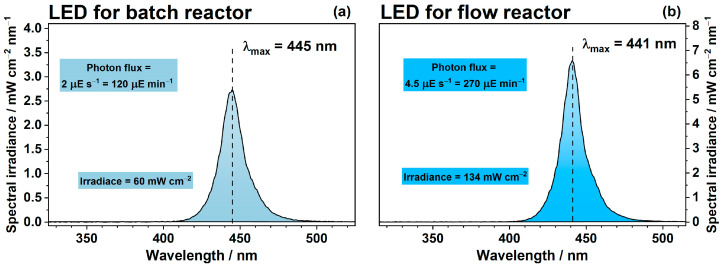
Emission spectra of LEDs used for the photocatalytic experiments in batch reactor (**a**) and in continuous-flow set-up (**b**). Colored rectangle areas show corresponding values of photon flux and specific irradiance of photocatalyst.

**Table 1 ijms-24-05693-t001:** Atomic ratios of the elements on the surface of Fe/Bi_2_WO_6_/TiO_2_-N before and after the stability test.

	O/Ti	N/Ti	Bi/Ti	W/Ti
Before reaction	3.99	0.10	0.60	0.42
After reaction	3.81	0.10	0.61	0.41

## Data Availability

The data presented in this study are available on request from the corresponding author. The data are not publicly available due to privacy.

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
