# Peer review of "Kinetic Aspects of Benzene Degradation over TiO_2_-N and Composite Fe/Bi_2_WO_6_/TiO_2_-N Photocatalysts under Irradiation with Visible Light"

_ijms, 2023, doi:10.3390/ijms24065693_

Round 1

Reviewer 1 Report

The present manuscript reports on the “Kinetic Aspects of Benzene Degradation over TiO2-N and Composite Fe/Bi2WO6/TiO2-N Photocatalysts under Irradiation with Visible Light”. The work is of some interest but seems to be too primitive and lacks novelty, proper scientific support and justification. The synthesized TiO2-N and Composite Fe/Bi2WO6/TiO2-N are not sufficiently characterized to support the claims. There are many reports exhibiting on catalysis. Thus, in my opinion, the manuscript in its present form cannot be considered for publication. I recommend major revision.

Following are some of the comments/suggestions which will be useful to the authors.

1. First of all, there are many previous works published for adsorption process. The authors seem deliberately avoid those papers. This is unusual, as the authors need to acknowledge the previous literature and compare their work with the similar ones in the literature and demonstrate their research outcomes in terms of advantages and disadvantages. Some of studies are given below need to cited at suitable place; doi.org/10.3390/catal13020231; DOI: 10.1039/D2RA07932A; doi.org/10.1016/j.jphotochem.2020.112776; doi.org/10.1016/j.molliq.2021.116270; DOI: 10.1039/D2RA01058E

2. Label the peaks of XRD spectra.

3. Correct the units of XRD spectra.

4. The morphology and size of the synthesized products are not cleared by SEM results. Therefore, TEM results are required.

5. Add EDX results to indicate the presence of elements in the synthesized product.

6. Improve the UV-Vis results of Figure 3.

7. Where are the XPS results?

8. Add the catalytic reaction mechanism.

9. Discussion portion is very weak in this manuscript. Improve it.

10. Improve the abstract by adding some results of this manuscript.

11. Improve the title of the manuscript.

Author Response

Reviewer #1

Comment: The present manuscript reports on the “Kinetic Aspects of Benzene Degradation over TiO2-N and Composite Fe/Bi2WO6/TiO2-N Photocatalysts under Irradiation with Visible Light”. The work is of some interest but seems to be too primitive and lacks novelty, proper scientific support and justification. The synthesized TiO2-N and Composite Fe/Bi2WO6/TiO2-N are not sufficiently characterized to support the claims. There are many reports exhibiting on catalysis. Thus, in my opinion, the manuscript in its present form cannot be considered for publication. I recommend major revision

Response: This manuscript, submitted to the special issue entitled "Molecular Aspects in Catalytic Materials for Pollution Elimination and Green Chemistry", considers the photocatalytic oxidation method for the degradation of volatile organic compounds. It is focused on studying the kinetic aspects of degradation for different types of pollutants (not on the design of new materials). In this point of view, this manuscript provides new information concerning the peculiarities in the degradation of aromatic compounds under visible light that you can’t find in the published papers. Another point is that the proposed multicomponent Fe/Bi2WO6/TiO2-N system is also new and poorly studied. We provide and discuss in the paper all important characteristics of the photocatalysts but mainly focused on the discussion of the kinetic aspects. Other detailed characteristics of the photocatalysts are referred to and discussed in our previously published papers cited in the manuscript.

Comment: Following are some of the comments/suggestions which will be useful to the authors. 1. First of all, there are many previous works published for adsorption process. The authors seem deliberately avoid those papers. This is unusual, as the authors need to acknowledge the previous literature and compare their work with the similar ones in the literature and demonstrate their research outcomes in terms of advantages and disadvantages. Some of studies are given below need to cited at suitable place; doi.org/10.3390/catal13020231; DOI: 10.1039/D2RA07932A; doi.org/10.1016/j.jphotochem.2020.112776; doi.org/10.1016/j.molliq.2021.116270; DOI: 10.1039/D2RA01058E.

Response: This study is focused on the photocatalytic oxidation method for the degradation of volatile organic compounds. Adsorption is the essential step of this photocatalytic process, and it is considered and discussed in many papers cited in the Introduction and the main text of manuscript. Adsorption process is pointed in the figures and discussed in detail thorough whole text of manuscript for all cases. Furthermore, as stated in the conclusions, the adverse effect observed for benzene is due to occupation of available adsorption sites on the surface of the photocatalyst by non-volatile intermediates. Therefore, we disagree to the opinion of the reviewer that the adsorption process is avoided in this paper.

Concerning the proposed papers:

[1] M. Arif, U. Fatima, A. Rauf, Z.H. Farooqi, M. Javed, M. Faizan, S. Zaman, A New 2D Metal-Organic Framework for Photocatalytic Degradation of Organic Dyes in Water, Catalysts. 13 (2023) 231. https://doi.org/10.3390/catal13020231.

[2] M. Arif, Catalytic degradation of azo dyes by bimetallic nanoparticles loaded in smart polymer microgels, RSC Adv. 13 (2023) 3008–3019. https://doi.org/10.1039/D2RA07932A.

[3] M.A. Qamar, S. Shahid, M. Javed, S. Iqbal, M. Sher, M.B. Akbar, Highly efficient g-C3N4/Cr-ZnO nanocomposites with superior photocatalytic and antibacterial activity, Journal of Photochemistry and Photobiology A: Chemistry. 401 (2020) 112776. https://doi.org/10.1016/j.jphotochem.2020.112776.

[4] M. Arif, Z.H. Farooqi, A. Irfan, R. Begum, Gold nanoparticles and polymer microgels: Last five years of their happy and successful marriage, Journal of Molecular Liquids. 336 (2021) 116270. https://doi.org/10.1016/j.molliq.2021.116270.

[5] M. Arif, Complete life of cobalt nanoparticles loaded into cross-linked organic polymers: a review, RSC Adv. 12 (2022) 15447–15460. https://doi.org/10.1039/D2RA01058E.

we have cited ref#3 but other papers are not relevant to the topic of the submitted manuscript.

Comment: 2. Label the peaks of XRD spectra

Response: According to your recommendation, we have labeled the peaks of XRD patterns.

Comment: 3. Correct the units of XRD spectra.

Response: In general view, the presented units in XRD patterns are correct. To be more correct, we changed plural form of units (“degrees”) to a single form (“degree”).

Comment: 4. The morphology and size of the synthesized products are not cleared by SEM results. Therefore, TEM results are required.

Response: According to your recommendation, we have added TEM micrographs of Bi2WO6/TiO2-N obtained using HAADF imaging and EDX mapping techniques. The information on the size of crystallites was also added.

Comment: 5. Add EDX results to indicate the presence of elements in the synthesized product.

Response: According to your recommendation, we have added TEM micrographs obtained using EDX mapping technique to show element mapping.

Comment: 6. Improve the UV-Vis results of Figure 3.

Response: Figure 3 was improved.

Comment: 7. Where are the XPS results?.

Response: The results of XPS analysis are shown in Figure 2 and Figure S3. In the revised version of manuscript, we have added the results of XPS analysis of the samples after stability test (Figure 9, Table 1, Figure S7, Table S1). Please find the revised version of manuscript.

Comment: 8. Add the catalytic reaction mechanism.?.

Response: According to your recommendation and the recommendation of reviewer#2, we more discussed the mechanism of benzene degradation in the text of manuscript. Please find the revised version of manuscript.

Comment: 9. Discussion portion is very weak in this manuscript. Improve it.

Response: We improved the discussion of the results in the main text of manuscript. Please find the revised version of manuscript.

Comment: 10. Improve the abstract by adding some results of this manuscript

Response: We have corrected the Abstract. Please find the revised version of manuscript.

Comment: 11. Improve the title of the manuscript.

Response: The current title well corresponds to the topic of manuscript.

Reviewer 2 Report

In this manuscript, the authors reported the synthesis and characterization of TiO2-N and Fe/Bi2WO6/TiO2-N catalysts for the degradation of volatile organic compounds under the visible light. It can be accepted after major revision before this manuscript should be polished and modified further. Please see some comments as follows.

1.     In the section of Introduction, it is long-winded. It should be rewritten.

2.     The surface state of the catalyst is very important. The XPS of TiO2-N and composite Fe/Bi2WO6/TiO2-N catalyst after the reaction should be added into the revised manuscript.

3.     What is the mechanism of the benzene degradation over TiO2-N and composite Fe/Bi2WO6/TiO2-N photocatalysts?

4.     It is valuable to provide the activity data of commercial degussa TiO2 (P25). It is a standard photocatalyst used to evaluate the activity of different catalysts.

5.     What is the real active site of TiO2-N and Fe/Bi2WO6/TiO2-N catalysts for the benzene degradation?

6.     The stability is very important for the catalysts. It should be added the stability test.

Author Response

Reviewer #2

Comment: In this manuscript, the authors reported the synthesis and characterization of TiO2-N and Fe/Bi2WO6/TiO2-N catalysts for the degradation of volatile organic compounds under the visible light. It can be accepted after major revision before this manuscript should be polished and modified further. Please see some comments as follows.

Response: Thank you for your appreciation of our work and your valuable comments helping us to improve this paper. A detailed response to each comment is provided below.

Comment: 1. In the section of Introduction, it is long-winded. It should be rewritten.

Response: We have rewritten and shorten the text in Introduction according to your recommendation. Please find the revised version of this section.

Comment: 2. The surface state of the catalyst is very important. The XPS of TiO2-N and composite Fe/Bi2WO6/TiO2-N catalyst after the reaction should be added into the revised manuscript.

Response: According to your recommendation, we have performed the long-term stability test and made XPS analysis of the photocatalysts before and after this test. The XPS results were added the main text of manuscript and the Supplementary Materials. Please find the revised version of manuscript.

Comment: 3. What is the mechanism of the benzene degradation over TiO2-N and composite Fe/Bi2WO6/TiO2-N photocatalysts?

Response: Both studied systems are similar because the key component, which provides a high level of visible-light activity in the degradation of VOCs, is TiO2-N. Therefore, there was no reason to find differences in the mechanism of benzene degradation over these systems. We relied on the published papers concerning the mechanism of benzene degradation in detail:

According to the literature data [1–3], the holes (h+) photogenerated in the conduction band of the photocatalyst can directly react with adsorbed benzene molecules to form phenyl radical cations (C6H5•+), as well as oxidize adsorbed H2O molecules to form OH radicals. C6H5•+ can further react with the adsorbed O2 or O2•- to form a peroxide radical and, sequentially, phenol [6]. An alternative pathway involves the direct interaction of C6H5•+ with OH to form phenol and other hydroxylated intermediates (e.g., hydroquinone and benzoquinone) [7]. These compounds can be completely oxidized to CO2 and H2O through a series of steps that include opening the aromatic ring and oxidation of non-cyclic hydrocarbons. In parallel pathways, C6H5•+ can interact with other adsorbed benzene molecules to form carbon deposits due to polymerization. Accumulation of these deposits can substantially reduce the ability of photocatalyst to oxidize benzene molecules due to blocking its adsorption sites. Other transformation routes [8] include the reactions of photoinduced OH with adsorbed benzene molecules to form various types of alkyl radicals (e.g., CH3, C2H5 and C6H5), which also contribute to the overall decomposition of benzene [9].

We more discussed this mechanism of benzene degradation in the text of manuscript. Please find the revised version of manuscript.

  1. d’Hennezel, O.; Pichat, P.; Ollis, D.F. Benzene and Toluene Gas-Phase Photocatalytic Degradation over H2O and HCL Pretreated TiO2: By-Products and Mechanisms. Journal of Photochemistry and Photobiology A: Chemistry1998, 118, 197–204, doi:10.1016/S1010-6030(98)00366-9.
  2. Einaga, H.; Futamura, S.; Ibusuki, T. Photocatalytic Decomposition of Benzene over TiO2 in a Humidified Airstream. Phys. Chem. Chem. Phys. 1999, 1, 4903–4908, doi:10.1039/a906214i.
  3. Einaga, H. Heterogeneous Photocatalytic Oxidation of Benzene, Toluene, Cyclohexene and Cyclohexane in Humidified Air: Comparison of Decomposition Behavior on Photoirradiated TiO2 Catalyst. Applied Catalysis B: Environmental 2002, 38, 215–225, doi:10.1016/S0926-3373(02)00056-5.
  4. Kozlov, D.V. Titanium Dioxide in Gas-Phase Photocatalytic Oxidation of Aromatic and Heteroatom Organic Substances: Deactivation and Reactivation of Photocatalyst. Theor Exp Chem 2014, 50, 133–154, doi:10.1007/s11237-014-9358-6.
  5. Bui, T.D.; Kimura, A.; Higashida, S.; Ikeda, S.; Matsumura, M. Two Routes for Mineralizing Benzene by TiO2-Photocatalyzed Reaction. Applied Catalysis B: Environmental 2011, 107, 119–127, doi:10.1016/j.apcatb.2011.07.004.
  6. Zhuang, H.; Gu, Q.; Long, J.; Lin, H.; Lin, H.; Wang, X. Visible Light-Driven Decomposition of Gaseous Benzene on Robust Sn 2+ -Doped Anatase TiO 2 Nanoparticles. RSC Adv. 2014, 4, 34315–34324, doi:10.1039/C4RA05904B.
  7. Vikrant, K.; Park, C.M.; Kim, K.-H.; Kumar, S.; Jeon, E.-C. Recent Advancements in Photocatalyst-Based Platforms for the Destruction of Gaseous Benzene: Performance Evaluation of Different Modes of Photocatalytic Operations and against Adsorption Techniques. J. Photochem. Photobiol., C 2019, 41, 100316, doi:10.1016/j.jphotochemrev.2019.08.003.
  8. Bathla, A.; Vikrant, K.; Kukkar, D.; Kim, K.-H. Photocatalytic Degradation of Gaseous Benzene Using Metal Oxide Nanocomposites. Advances in Colloid and Interface Science 2022, 305, 102696, doi:10.1016/j.cis.2022.102696.
  9. He, F.; Ma, F.; Li, T.; Li, G. Solvothermal Synthesis of N-Doped TiO2 Nanoparticles Using Different Nitrogen Sources, and Their Photocatalytic Activity for Degradation of Benzene. Chinese Journal of Catalysis 2013, 34, 2263–2270, doi:10.1016/S1872-2067(12)60722-0.

Comment: 4. It is valuable to provide the activity data of commercial degussa TiO2 (P25). It is a standard photocatalyst used to evaluate the activity of different catalysts.

Response: In contrast to the UV region, TiO2 P25 commonly has extremely low activity under blue light because it can absorb only small portion of photons. According to your recommendation, we have added the data on corresponding activity of TiO2 P25 under the same conditions as for other studied photocatalysts. Please find the revised version of manuscript.

Comment: 5. What is the real active site of TiO2-N and Fe/Bi2WO6/TiO2-N catalysts for the benzene degradation?

Response: Commonly, “active site” is an abstract concept for the photocatalysts because under irradiation many reactive species (photoinduced charge carriers and some radicals) are present on the surface of photocatalyst. After adsorption of pollutant (e.g., benzene) on the surface of photocatalyst, its molecules can participate in the redox transformations with the reactive species. Therefore, the number of adsorption sites for pollutant molecules can be used as an upward evaluation for the number of active sites. However, the efficiency of interfacial charge transfer commonly has more drastic effect on the photocatalytic rate.

Concerning predominant type of reactive species, in the case of TiO2-mediated photocatalytic oxidation under humidified air conditions, OH-radicals are regarded as the main species that provide oxidation reactions. Also, as mentioned before in discussing the mechanism, the photogenerated holes are important in the case of benzene degradation.

We more discussed this aspect in the text of manuscript. Please find the revised version of manuscript.

Comment: 6. The stability is very important for the catalysts. It should be added the stability test.

Response: We have performed long-term test on stability of photocatalysts in the degradation of benzene vapor and added the results of these experiments to the manuscript. Please find the revised version of manuscript.

Comment: Extensive editing of English language and style required.

Response: According to your recommendation, English language in the manuscript was thoroughly checked and edited.

The revised version of manuscript was checked using AJE Grammar Check tool (https://www.aje.com/grammar-check/). This tool shows 8.6/10 score (95st percentile of papers submitted to AJE) and indicates that the manuscript is well written and does not need extensive language editing. Final polishing will be made at proof-reading step.

Reviewer 3 Report

The authors have presented the work "Kinetic Aspects of Benzene Degradation over TiO2-N and Composite Fe/Bi2WO6/TiO2-N Photocatalysts under Irradiation with Visible Light". This manuscript has compared the photocatalytic performances of these composite materials in the photodegradation of ethanol and benzene vapor as well as giving sufficient evidence for the successful modification of the composite materials. The kinetic aspects of the degradation of volatile organic compounds are also well discussed and therefore, this manuscript can be published with minor revisions as shown below:

1. In p. 7 line 235-236, it is stated that the formation of formaldehyde in the gas phase was not observed in Figure S3. It is suggested that the line and caption for formaldehyde in Figure S3 should be shown so that readers will know about the full intermediate products that can be observed in PCO of ethanol.

2. It is suggested to include arrows indicating which concentration of the chemical is referred to (i.e. left y-axis or right y-axis) for Figure S3 and S4 in the Supplementary Materials so that it is consistent with the arrows as shown in Figure 4 in the main paper.

Author Response

Responses to reviewer’s comments

Reviewer #3

Comment: The authors have presented the work "Kinetic Aspects of Benzene Degradation over TiO2-N and Composite Fe/Bi2WO6/TiO2-N Photocatalysts under Irradiation with Visible Light". This manuscript has compared the photocatalytic performances of these composite materials in the photodegradation of ethanol and benzene vapor as well as giving sufficient evidence for the successful modification of the composite materials. The kinetic aspects of the degradation of volatile organic compounds are also well discussed and therefore, this manuscript can be published with minor revisions as shown below.

Response: Thank you for your appreciation of our work and your valuable comments helping us to improve this paper. A detailed response to each comment is provided below.

Comment: 1. In p. 7 line 235-236, it is stated that the formation of formaldehyde in the gas phase was not observed in Figure S3. It is suggested that the line and caption for formaldehyde in Figure S3 should be shown so that readers will know about the full intermediate products that can be observed in PCO of ethanol.

Response: We have improved Figure S3 according to your recommendation. Please find the revised version of this figure.

Comment: 2. It is suggested to include arrows indicating which concentration of the chemical is referred to (i.e. left y-axis or right y-axis) for Figure S3 and S4 in the Supplementary Materials so that it is consistent with the arrows as shown in Figure 4 in the main paper.

Response: We have improved Figures S3 and S4 according to your recommendation. Please find the revised version of these figures.

Round 2

Reviewer 1 Report

Accept

Reviewer 2 Report

It can be accepted.